# Factors Influencing Mammographic Density in Asian Women: A Retrospective Cohort Study in the Northeast Region of Peninsular Malaysia

**DOI:** 10.3390/diagnostics12040860

**Published:** 2022-03-30

**Authors:** Tengku Muhammad Hanis, Wan Nor Arifin, Juhara Haron, Wan Faiziah Wan Abdul Rahman, Nur Intan Raihana Ruhaiyem, Rosni Abdullah, Kamarul Imran Musa

**Affiliations:** 1Department of Community Medicine, School of Medical Sciences, Universiti Sains Malaysia, Kubang Kerian 16150, Kelantan, Malaysia; tengkuhanismokhtar@gmail.com; 2Biostatistics and Research Methodology Unit, School of Medical Sciences, Universiti Sains Malaysia, Kubang Kerian 16150, Kelantan, Malaysia; wnarifin@usm.my; 3Department of Radiology, School of Medical Sciences, Universiti Sains Malaysia, Kubang Kerian 16150, Kelantan, Malaysia; drjuhara@usm.my; 4Breast Cancer Awareness and Research Unit, Hospital Universiti Sains Malaysia, Kubang Kerian 16150, Kelantan, Malaysia; wfaiziah@usm.my; 5Department of Pathology, School of Medical Sciences, Universiti Sains Malaysia, Kubang Kerian 16150, Kelantan, Malaysia; 6School of Computer Sciences, Universiti Sains Malaysia, USM 11800, Penang, Malaysia; intanraihana@usm.my (N.I.R.R.); rosni@usm.my (R.A.)

**Keywords:** breast density, breast cancer, risk factors, Asians, Malaysia

## Abstract

Mammographic density is a significant risk factor for breast cancer. In this study, we identified the risk factors of mammographic density in Asian women and quantified the impact of breast density on the severity of breast cancer. We collected data from Hospital Universiti Sains Malaysia, a research- and university-based hospital located in Kelantan, Malaysia. Multivariable logistic regression was performed to analyse the data. Five significant factors were found to be associated with mammographic density: age (OR: 0.94; 95% CI: 0.92, 0.96), number of children (OR: 0.88; 95% CI: 0.81, 0.96), body mass index (OR: 0.88; 95% CI: 0.85, 0.92), menopause status (yes vs. no, OR: 0.59; 95% CI: 0.42, 0.82), and BI-RADS classification (2 vs. 1, OR: 1.87; 95% CI: 1.22, 2.84; 3 vs. 1, OR: 3.25; 95% CI: 1.86, 5.66; 4 vs. 1, OR: 3.75; 95% CI: 1.88, 7.46; 5 vs. 1, OR: 2.46; 95% CI: 1.21, 5.02; 6 vs. 1, OR: 2.50; 95% CI: 0.65, 9.56). Similarly, the average predicted probabilities were higher among BI-RADS 3 and 4 classified women. Understanding mammographic density and its influencing factors aids in accurately assessing and screening dense breast women.

## 1. Introduction

Breast cancer is the most common cancer among women in several Asian countries such as Japan, Singapore, Malaysia, Indonesia, China, South Korea, and Iran [1,2,3,4,5,6]. An increased incidence had been observed in countries such as Malaysia, China, India, and Thailand [5,6,7]. Thus, identifying significant factors associated with breast cancer is important in the management and prevention of the disease. Breast cancer can be defined as any type of abnormal growth of the breast cells. Histologically, the most common subtypes of breast cancer are ductal carcinoma and lobular carcinoma [8]. Breast cancer is a multifactorial disease. Several risk factors known to increase the risk of breast cancer are the late age of menopause, hormonal contraception, family history, and lifestyle-related factors such as alcohol consumption, smoking, and physical inactivity [9,10].

Another critical risk factor of breast cancer is mammographic density [11,12]. Mammographic density indicates a mitotic activity of the breast cells and their susceptibility to genetic damage, both of which influence breast cancer development [13]. In addition, susceptibility to genetic damage is affected by a mutagen, some hormones, and growth factors. Mammographic density had been observed to be higher in the Asian population or individuals with Asian ancestry compared to other populations [14,15]. The risk of breast cancer in dense breast women was higher than in non-dense breast women up to four to sixtimes [16,17]. Additionally, mammographic density makes a visual diagnosis, especially via mammogram, more challenging. Mammographic density reduces the accuracy and sensitivity of mammograms in breast cancer detection through the masking effect. Fibroglandular tissue appears white on the mammogram, while fatty tissue appears black. Therefore, accurate visual differentiation between the white-coloured fibroglandular tissue and the white-coloured cancer tissue becomes difficult, even among the experts. There are a few measures developed for categorising mammographic density. Quantitative measures such as Boyd classification and volumetric methods consider the quantity. In contrast, qualitative measures such as Wolfe classification, Tabar classification, and breast imaging-reporting and data system (BI-RADS) breast composition consider the quantity and density characteristics [18]. BI-RADS breast composition density ranges from almost entirely fatty tissue (BI-RADS A) to highly dense with no minimal fatty tissue (BI-RADS D).

Establishing significant factors influencing mammographic density is vital, as it aids in proper screening, follow-up, and surveillance of dense breast women. Furthermore, mammograms have been used in machine learning and deep learning studies [19]. Therefore, the information on the important risk factors of mammographic density will benefit future predictive studies. In this study, we aimed to determine factors affecting mammographic density in the northeast region of peninsular Malaysia and further quantified the relationship between breast density and the severity of breast cancer.

## 2. Materials and Methods

### 2.1. Study Site and Population

Malaysia has two regions: (1) West Malaysia, located on the Malay peninsula, and (2) East Malaysia, located on Borneo’s island. Furthermore, the Malaysia peninsula region is divided into thirteen states and three federal territories. The northeast region of peninsular Malaysia comprises three states: Kelantan, Terengganu, and Pahang. The three states share relatively similar demographic characteristics of their population. This retrospective study was conducted in Kelantan, mainly in Hospital Universiti Sains Malaysia (HUSM). HUSM is a research- and university-based hospital located inside the Universiti Sains Malaysia, Kelantan. This study received ethical approval from the human research ethics committee of Universiti Sains Malaysia (JEPeM) (USM/JEPeM/19090536). This study used secondary data. Thus, patients’ consents were not applicable.

### 2.2. Breast Cancer Data

Breast cancer data were collected from two departments and a unit in HUSM: the departments of Pathology and Radiology, and the Breast Cancer Awareness and Research Unit (BestARi). The information on the final diagnosis of breast cancer patients was collected from the pathology department. In contrast, the information on BI-RADS breast composition density and BI-RADS classification was collected from the radiology department. Sociodemographic data collected from medical records from the BestARi unit include the following: age (at the time of visit to BestARi unit), age of menarche, number of children, gender, race, weight, height, menopause status, family history of breast cancer, and history of birth control, hormone replacement, andtotal abdominal hysterectomy bilateral salpingo-oophorectomy (TAHBSO). Additionally, a body mass index (BMI) variable was calculated using the weight and height variables. All the data were limited to 1 January 2014 and 30 June 2021. Subsequently, the three datasets were combined into one dataset.

### 2.3. Study Design and Patient Selection

This retrospective cohort study included women who attended the BestARi unit, thus excluding non-HUSM or referral patients from other hospitals. Referral patients may be breast cancer patients referred to HUSM for further treatment or diagnosis. Breast cancer patients without BI-RADS breast composition density, missing BI-RADS classification, or with a BI-RADS classification of zero were excluded from the study. A final diagnosis of normal, benign, or malignant breast cancer was determined using histopathological examination (HPE) results. BI-RADS classification was used in the final diagnosis of breast cancer patients with missing HPE results. BI-RADS 1 was classified as normal, BI-RADS 2 and 3 were classified as benign, and BI-RADS 4, 5, and 6 were classified as malignant. BI-RADS breast composition density was further recategorised into non-dense and dense breast classification. BI-RADS A and B were classified as non-dense, while BI-RADS C and D were classified as dense.

### 2.4. Statistical Analysis and Software

R software ran data wrangling, exploration, and statistical analysis [20]. Data exploration was done to identify the duplicates and percentage of missing data. The latest medical record was taken for patients with several medical records.

#### 2.4.1. Missing Data Handling

Missing data types were determined using Little’s test [21]. This test is available in the naniar package [22]. *p*-value > 0.05 indicates the missingness is missing completely at random; otherwise, the type of missingness is missing at random or missing not at random. Since *p*-value was < 0.05 for the Little’s test in our data, missing values were imputed using multiple imputations. The mice package was used to run a fully specified conditional model or multivariable imputation by chained equation, a type of multiple imputation approach [23]. The number of imputations was set to 40. The convergence of the algorithm was evaluated with an iteration set to 45. A predictive mean matching model was used for numerical variables with missing data, binary logistic regression was used for two-level categorical variables, and polytomous logistic regression was used for more than two-level categorical variables.

#### 2.4.2. Logistic Regression

Subsequently, binary logistic regression was run on 40 imputed datasets. The parameter estimates of each result were pooled into a final result using Rubin’s rule, which was already applied in the mice package [24]. For each variable, a univariable analysis was done. Variables with *p*-values < 0.25 were included in the multivariable analysis. The most nonsignificant variable (*p*-values > 0.05) in the multivariable logistic regression was excluded at a time. A multivariate Wald test was used to compare a model with and without the variable. *p*-values < 0.05 indicated the variable was important, and thus, it was retained in the model. This backward selection process ended if all variables in the model demonstrated *p*-values < 0.05 or the model comparison demonstrated a *p*-value < 0.05.

The final model was checked for two-way interactions individually. *p*-values < 0.05 indicated a significant interaction and were retained in the model. Next, the model was evaluated for multicollinearity using a generalised variance inflation factor (GVIF) [25] available in the car package [26]. GVIF values above 10 for any variable indicated multicollinearity. Model fitness was evaluated using a classification table, area under the curve (AUC), and Hosmer–Lemeshow goodness of fit test [27]. A value above 70% for the classification table, above 70% for the AUC, and *p*-values > 0.05 indicated a model fit. Lastly, the linear relationship of the numerical variable and the logit of the outcome was evaluated using the Box and Tidwell method [28]. A *p*-value > 0.05 indicated a linear relationship.

Average predicted probabilities from the final model were used to quantify the relationship between breast density and the severity of breast cancer. This second objective was calculated only if the BI-RADS classification or diagnosis variables that reflected the cancer’s severity were significant. First, the predicted probabilities were calculated for each patient. Then, the mean and the standard deviation of the probabilities were calculated for each BI-RADS classification.

## 3. Results

After removing individuals with a BI-RADS classification of 0, the remaining 1091 women were included in the analysis. There were 54.3% non-dense breast women and 45.7% dense breast women in the data. The three main races in Malaysia are Malay, Chinese, and Indian. However, there were only four Indian women in the data. Thus, they were grouped in with others. Most women in our data, regardless of mammographic density, were Malay, presented with no family history of breast cancer, presented with no history of birth control, hormone replacement, or TAHBSO, were classified as BI-RADS 2, and were diagnosed with benign breast cancer. The main difference between the non-dense and dense breast women was their menopause status. The non-dense group consisted of postmenopausal women, while the dense group consisted mainly of pre-menopausal women. The detailed characteristics of women who attended the BestARi unit, HUSM, are presented in Table 1.

The imputation models were checked for the algorithm’s convergence once multiple imputations were done to impute the missing values. The convergent plots presented in the Appendix A show the algorithm converged. The result of the univariable analysis on the imputed data is presented in Table 2.

Table 3 presents the final multivariable logistic regression model. There were no significant two-way interactions or multicollinearity among the predictors. Additionally, no linear relationship was found between the numerical variables and the logit outcomes. Therefore, the model was considered fit; the model’s goodness of fit details are presented in the Appendix A. In this study, there were five significant factors of mammographic density identified: age, number of children, BMI, menopause status, and BI-RADS classification. Subsequently, the average predicted probabilities of dense breasts were calculated from the model in Table 3 for each BI-RADS classification (Figure 1). Women with BI-RADS 4 demonstrated the highest probabilities of presenting with dense breasts, while women with BI-RADS 1 demonstrated the lowest probabilities of presenting with dense breasts.

## 4. Discussion

In this study, we determined factors affecting mammographic density in the northeast region of peninsular Malaysia and quantified the relationship between breast density and the severity of breast cancer. Five significant factors were found to be associated with mammographic density: age, number of children, BMI, menopause status, and BI-RADS classification. The average predicted probabilities of having a dense breast were higher among BI-RADS 3 and 4 classified women.

This study found an inverse relationship between age, number of children, menopause status, and mammographic density, with the highest effect observed in the menopause status. As age increased, the women tended to have more children and went through menopause. However, no significant two-way interaction was found between these variables in our model. Several studies found a similar trend of the inverse relationship between age and menopause status and mammographic density [29,30,31,32,33]. Mammographic density decreased as the women got older, especially around the menopausal age [34]. About 63.7% of the non-dense group were postmenopausal women, and the dense group made up 61.9% of pre-menopausal women in this study (Table 1). Additionally, lack of parity was a significant risk factor for a dense breast among women, as found in the previous studies [35,36]. Lobular involution had been observed to be lower in parous women, reducing the degree of mammographic density, especially in pre-menopausal women [37,38,39].

Furthermore, researchers found that women with a higher BMI were less likely to present with dense breasts. Several studies found a similar finding [36,40,41,42]. Additionally, BMI and physical activity had been recognised as the determinants of mammographic density change [43,44]. However, other studies did not find any significant relationship between physical activity and mammographic density [45,46]. Unfortunately, physical activity and other lifestyle-related variables such as dietary intake and smoking were not available in our data. Some of these variables, such as alcohol consumption, had been observed to influence the mammographic density [47,48].

The average predicted probabilities of mammographic density were highest in BI-RADS 3 and 4 (Figure 1). Similarly, the highest odds ratio was observed among BI-RADS 4 vs. BI-RADS 1 at 3.75, followed by BI-RADS 3 vs. BI-RADS 1 at 3.25 (Table 3). Thus, BI-RADS 3 and 4 classified women need a careful assessment, as this classification reflected a very suspicious lesion that warrants further imaging investigations such as supplemented ultrasound or MRI. Furthermore, the high mammographic density in women with BI-RADS 3 and 4 further complicates the assessment, as the dense breast reduces mammograms’ accuracy and sensitivity in breast cancer detection. Subsequently, the dense breast may prevent an early lesion diagnosis or, worse, lead to a misdiagnosis.

There were a few limitations to our study. This study used secondary data collected from two different departments and a unit in HUSM. Thus, the variables included were limited to pre-existing variables from primary sources. Additionally, there were substantial missing values in our data. Although multiple imputations had been observed to obtain an unbiased result even in data with up to 90% missing values [49], the possibility of biased results is still present, mainly in the case that the imputation models are misspecified. Future studies in Malaysia should try to prevent missing data in their studies. Further studies should also explore the effect of modifiable risk factors such as physical activity, diet, smoking, and educational attainment concerning mammographic density and breast cancer risk in the Malaysian population setting.

## 5. Conclusions

Mammographic density is a significant predictor of breast cancer risk. Therefore, a better understanding of factors influencing mammographic density helps plan an accurate assessment of breast cancer patients. In this study, we identified five factors affecting mammographic density among women in the northeast region of peninsular Malaysia. Our findings may be generalised to neighbouring regions with relatively similar population characteristics.

## Figures and Tables

**Figure 1 diagnostics-12-00860-f001:**
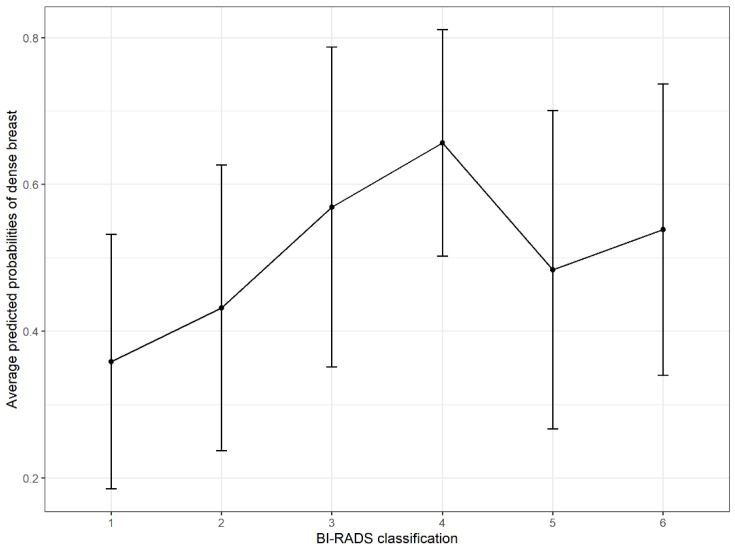
Average predicted probabilities of dense breast women who attended BestARi unit in Hospital Universiti Sains Malaysia according to BI-RADS classification. The point reflects the average predicted probabilities of being a dense breast woman and the length of the error bars reflects the standard deviation of the probabilities.

**Table 1 diagnostics-12-00860-t001:** Characteristics of women who attended BestARi unit in Hospital Universiti Sains Malaysia (*n* = 1091).

Variables	Non-Dense n (%)	Dense n (%)	Missing Values n (%)
Age (years) ^1^	55.5 (9.6)	49.9 (8.3)	3 (0.3)
Age at menarche (years) ^1^	13.1 (1.5)	13.1 (1.5)	97 (8.9)
Weight (kg) ^1^	65.9 (12.7)	60.9 (12.3)	263 (24.0)
Height (cm) ^1^	155.2 (6.1)	155.3 (6.5)	692 (63.0)
Body mass index ^1^	27.7 (5.7)	25.6 (5.2)	696 (64.0)
Number of children ^1^	4 (2.5)	3 (2.6)	529 (48.0)
Race			34 (3.1)
Others	9 (1.5)	8 (1.7)	
Chinese	56 (9.6)	77 (16.2)	
Malay	517 (88.8)	390 (82.1)	
Menopause status			0 (0.0)
No	215 (36.3)	309 (61.9)	
Yes	377 (63.7)	190 (38.1)	
Family history			520 (48.0)
No	249 (81.1)	204 (77.3)	
Yes	58 (18.9)	60 (22.7)	
BC-HR			51 (4.7)
No	374 (65.7)	310 (65.8)	
Yes	195 (34.3)	161 (34.2)	
TAHBSO			70 (6.4)
No	498 (89.1)	409 (88.5)	
Yes	61 (10.9)	53 (11.5)	
BI-RADS classification			0 (0.0)
1	93 (15.7)	52 (10.4)	
2	379 (64.0)	288 (57.7)	
3	59 (10.0)	78 (15.6)	
4	23 (3.9)	44 (8.8)	
5	32 (5.4)	30 (6.0)	
6	6 (1.0)	7 (1.4)	
Diagnosis			0 (0.0)
Normal	124 (20.9)	106 (21.2)	
Benign	444 (75.0)	368 (73.7)	
Malignant	24 (4.1)	25 (5.0)	

Notes: BestARi = breast cancer awareness and research unit; Family history = family history of breast cancer; BC-HR = history of birth control or hormone replacement; TAHBSO = history of total abdominal hysterectomy bilateral salpingo-oophorectomy; ^1^ mean (SD).

**Table 2 diagnostics-12-00860-t002:** Univariable logistic regression of mammographic density of women who attended BestARi unit in Hospital Universiti Sains Malaysia (*m* = 40).

Variables	OR	95% (CI)	*p*-Value
Age	0.93	0.92, 0.95	<0.001
Age at menarche	1.02	0.94, 1.11	0.643
Weight	0.97	0.95, 0.98	<0.001
Height	1.01	0.99, 1.04	0.313
Body mass index	0.91	0.88, 0.94	<0.001
Number of children	0.88	0.82, 0.94	<0.001
Race			
Others	-	-	
Chinese	1.57	0.57, 4.35	0.383
Malay	0.84	0.32, 2.20	0.720
Menopause status			
No	-	-	
Yes	0.35	0.27, 0.45	<0.001
Family history			
No	-	-	
Yes	1.26	0.87, 1.83	0.227
BC-HR			
No	-	-	
Yes	0.98	0.76, 1.27	0.880
TAHBSO			
No	-	-	
Yes	1.04	0.71, 1.54	0.831
BI-RADS classification			
1	-	-	
2	1.36	0.94, 1.97	0.107
3	2.36	1.46, 3.82	<0.001
4	3.42	1.86, 6.29	<0.001
5	1.68	0.92, 3.07	0.093
6	2.09	0.67, 6.55	0.207
Diagnosis			
Normal	-	-	
Benign	0.97	0.72, 1.30	0.837
Malignant	1.22	0.66, 2.26	0.530

Notes: OR = Odds ratio; CI = Confidence interval; Family history = Family history of breast cancer; BC-HR = history of birth control or hormone replacement; TAHBSO = history of total abdominal hysterectomy bilateral salpingo-oophorectomy.

**Table 3 diagnostics-12-00860-t003:** Multivariable logistic regression of mammographic density of women who attended BestARi unit in Hospital Universiti Sains Malaysia (*m* = 40).

Variables	OR	95% (CI)	*p*-Value
Age	0.94	0.92, 0.96	<0.001
Number of children	0.88	0.81, 0.96	0.003
Body mass index	0.88	0.85, 0.92	<0.001
Menopause status			
No	-	-	
Yes	0.59	0.42, 0.82	0.002
BI-RADS classification			
1	-	-	
2	1.87	1.22, 2.84	0.004
3	3.25	1.86, 5.66	<0.001
4	3.75	1.88, 7.46	<0.001
5	2.46	1.21, 5.02	0.013
6	2.50	0.65, 9.56	0.180

Notes: OR = Odds ratio; CI = Confidence interval.

## Data Availability

The data are available upon reasonable request to the corresponding author.

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
