# Peer review of "Factors Influencing Mammographic Density in Asian Women: A Retrospective Cohort Study in the Northeast Region of Peninsular Malaysia"

_diagnostics, 2022, doi:10.3390/diagnostics12040860_

Round 1

Reviewer 1 Report

Summary: A retrospective analysis to reveal the influential factors on dense breast in mammography. 

Comments:

1. There are several relevant articles that are proposed to be discussed in the Discussion section:
- https://www.ncbi.nlm.nih.gov/pmc/articles/PMC8284716/- - https://pubmed.ncbi.nlm.nih.gov/26075657/

-https://jamanetwork.com/journals/jamanetworkopen/fullarticle/2783508

- https://www.nature.com/articles/s41523-018-0055-9

2. Introduction (lines 35-36): "
 Breast cancer is the most common cancer among women in several Asian countries 35 like Japan, Singapore, Malaysia, and China". It is also the most common cancer among women in Iran, South Korea, and Indonesia:
Authors can use the following articles for this notion. 
- https://brief.land/ijcm/articles/107043.html

- http://journal.waocp.org/article_88587.html

- http://journal.waocp.org/article_89592.html

3. Discussion section: It is recommended authors put forward the pathophysiology behind the association between dense breast and breast cancer. 1. One explanation is through the following mechanism. The authors can use the following words in the Discussion section:
Recent evidence has put forward the association between cell energy depletion and cancer formation (PMID: 31823627). In order that, decrease in cellular ATP results in genomic instability that in turn results in cancer formation. On the other hand, dense breasts expose cells to a decrease in cell energy through nutrition division between more cells. This notion can explain the association between dense breasts and cancer formation.
2. Second explanation is through oxidative stress. In order that, breasts with dense tissue experience more oxidative stress ( https://breast-cancer-research.biomedcentral.com/articles/10.1186/bcr1831). Oxidative stress is a sign of increased mitochondrial activity. In addition, breast cancer with higher mitochondria content has a poor prognosis (https://www.ncbi.nlm.nih.gov/pmc/articles/PMC8375016/). This notion can be justified by recent evidence that mitochondria are essential for cancer cells to survive and evade the immune system (https://www.preprints.org/manuscript/202201.0171/v2).
It is recommended author mention these hypothetical links between dense breasts and cancer by citing the aforementioned articles.  

4. Results: It is recommended authors evaluate the association between dense breast and malignant diagnosis and the pathologic grade (if available) on biopsy. 

Reviewer 2 Report

The authors evaluated the risk factors of mammographic density in Asian women. Five significant factors have been found to be related with mammographic density; age, number of children, BMI , menopause status and BI-RADS classification.

Although the study is interesting and the results are well described, it needs some improvements to be published on Diagnostics.

  1. The authors should better describe what they mean with the term "breast cancer". I think they mean "breast carcinoma". However, mammary gland may develop several forms of neoplasms, including epithelial, mesenchymal and mixed ones. A brief description of the multiple breast malignancies must be added to the manuscript (please see: 10.32074/1591-951X-31-19)
  2. Currently known prognostic factors of breast carcinoma should be better described and reported (please see: 10.3389/fonc.2020.01519)

Round 2

Reviewer 2 Report

The paper has been improved and is now acceptable in this current form.